# The Effect of Homocysteine on the Secretion of Il-1β, Il-6, Il-10, Il-12 and RANTES by Peripheral Blood Mononuclear Cells—An In Vitro Study

**DOI:** 10.3390/molecules26216671

**Published:** 2021-11-04

**Authors:** Magdalena Borowska, Hanna Winiarska, Marzena Dworacka, Anna Wesołowska, Grzegorz Dworacki, Przemysław Łukasz Mikołajczak

**Affiliations:** 1Department of Pharmacology, Poznan University of Medical Sciences, 60-806 Poznan, Poland; hwiniar@ump.edu.pl (H.W.); mdworac@ump.edu.pl (M.D.); anna.wesolowska@ump.edu.pl (A.W.); przemmik@ump.edu.pl (P.Ł.M.); 2Department of Clinical Pathology, Poznan University of Medical Sciences, 60-355 Poznan, Poland; gdwrck@ump.edu.pl

**Keywords:** homocysteine, Il-1β, Il-6, Il-12, Il-10, RANTES

## Abstract

The contemporary theory of the inflammatory-immunological pathomechanism of atherosclerosis includes the participation of interleukin-1β (Il), Il-6, Il-10, Il-12, RANTES, and homocysteine in this process. The knowledge on the direct effect of hyperhomocysteinemia on inflammatory-state-related atherosclerosis is rather scarce. Our study is the first to account for the effects of homocysteine on the secretion of Il-10 and RANTES in vitro conditions. For this purpose, human mitogen-stimulated peripheral blood mononuclear cells (PBMNCs) were cultured in vitro and exposed to homocysteine at high concentrations. Subsequently, the concentrations of cytokines were assayed in the cell culture supernatant using flow cytofluorimetry. It has been shown that, in the presence of homocysteine, the secretion of IL-1, IL-6 and RANTES by PBMNCs was increased, whereas IL-10 concentration was significantly lower than that of the supernatant derived from a mitogen-stimulated cell culture without homocysteine. The secretion of Il-12 by PBMNCs exposed exclusively to mitogen, did not differ from homologous cells also treated with homocysteine. Therefore, in our opinion, high-concentration homocysteine affects the progression of atherosclerosis by increasing the secretion of proinflammatory cytokines secreted by PBMNCs, such as Il-1β, Il-6, RANTES, and by attenuating the secretion of Il-10.

## 1. Introduction

Progressive atherosclerosis is a major causative factor in ischemic heart disease, myocardial infarction, stroke and other atherosclerosis-related complications [1]. The dramatically high mortality due to the process of atherosclerosis has encouraged us to pursue detailed research on all aspects of the pathogenesis of this process. It is known that the pathogenesis of atherosclerosis, at both the level of its initiation and its progression, involves a chronic mild inflammatory process. The current concept of the development of vascular complications includes the participation of numerous cytokines, which are treated as important elements of the pathogenesis of inflammation [2]. It should be emphasized that an increased secretion of Il-1β, Il-6, Il-12 and RANTES chemokines and, at the same time, a reduced secretion of anti-inflammatory Il-10 due to hypercholesterolaemia, oxidative stress and hyperglycaemia, resulted in vascular wall damage and the progression of atherosclerosis [3,4,5]. The involvement of these mediators has already been observed in the early stages of atherogenesis, as well as in the progression of atherosclerosis [4,6,7].

The contemporary theory of the inflammatory-immunological pathomechanism of atherosclerosis includes the participation of not only numerous cytokines, but also homocysteine in this process at almost every stage of the formation of the atherosclerotic plaque [8]. The role of proatherogenic homocysteine and cytokines has already been well-documented in many publications. The correlation between homocysteine levels and the concentration and expression of some cytokines and pro-inflammatory and proatherogenic factors has been previously demonstrated: Il-6 [9], Il-1β, Il-8, Il-12 [10], Il-10 [11] and RANTES [12,13]. However, the evidence of a direct link between hyperhomocysteinemia and the generation of an immune response leading to the exacerbation or suppression of cytokine secretion is unclear. Therefore, in vitro studies were conducted to investigate the effect of homocysteine on the secretion of selected cytokines by peripheral blood mononuclear cells. To date, there have been no reports detailing the direct effect of homocysteine on the secretion of Il-10 and RANTES chemokines by PBMNCs. The concentration of homocysteine is a modifiable proatherogenic factor; therefore, the identification of a possible direct link between hiperhomocysteinemia and Il-1β, Il-6, Il-10, Il-12 and RANTES secretion by PBMNCs could be of particular importance, considering the constantly increasing number of patients suffering from civilization diseases closely related to atherosclerosis.

## 2. Results

It has been shown that the secretion of Il-1β, Il-6 and RANTES by mitogen-stimulated PVMNCs increases (*p* ≤ 0.05) in the presence of homocysteine (Figure 1, Figure 2 and Figure 3).

At the same time, the concentration of IL-10 in a supernatant derived from a mitogen-stimulated cell culture in the presence of homocysteine is significantly lower (*p* ≤ 0.05) than in a supernatant derived from a mitogen-stimulated cell culture without the addition of homocysteine to the medium (Figure 4).

The secretion of Il-12 by cells coming from healthy individuals exposed exclusively to mitogen did not differ from homologous cells that were also treated with homocysteine (Figure 5).

Figure 6, Figure 7, Figure 8 and Figure 9 demonstrate examples of cytometry graphs for patients that are representative of each examined group.

## 3. Discussion

Studies conducted in recent years have indicated a close relationship between atherosclerosis and immunologic processes in the vascular wall cells which were influenced by, e.g., interleukins, chemokines and adipocytokines [14,15,16], as well as hyperhomocysteinemia [17]. It is well known that proatherogenic Il-1β has the capacity to stimulate the chemokines, Il-1β, Il-2, Il-6 secretion and to activate the inducible nitric oxide synthase, which enhances the formation of reactive oxygen species [18]. Moreover, Il-1β and Il-6 have been found to enhance the migration and proliferation of Vascular Smooth Muscle Cells (VSMC) and the synthesis of adhesins such as Vascular Cell Adhesion Molecule 1 (VCAM-1) and Intracellular Cell Adhesion Molecule 1 (ICAM-1) [19,20]. Additionally, Il-6 is a major factor in the hepatocytes stimulation, inducing the C reactive protein (CRP) and fibrinogen secretion, which are recognized as the markers and components of inflammatory response occurring during atherogenesis [21,22]. The anti-inflammatory role of Il-10 includes the inhibition of the secretion of the pro-inflammatory Il-1β, Il-6, Il-8, Il-12 and TNF-α secretion by Th1, as well as the induction of Th2-type response, resulting in an increased secretion of Il-4, Il-5, Il-10, Il-13 [19]. It is noteworthy that Il-10 participates in the regulation of the expression of proinflammatory particles (Il-1β, Il-6, tissue factor, TF), adhesion molecules (VCAM-1, ICAM-1), and chemotactic factors (Il-8) [19]. Low levels of Il-10 are, therefore, associated with atherosclerosis and are considered to be one of the non-classical proatherogenic factors [23]. The role of RANTES in the atherosclerotic process consists of the involvement of this chemokine in the process of the recruitment of inflammatory cells, i.e., T cells, monocytes and macrophages to the vessel wall [24,25,26]. The effect is an increase in the vascular smooth muscle proliferation and the dynamics of the plaque formation, which have a favourable influence on its stabilization [27,28]. The mechanism by which homocysteine promotes atherosclerosis includes the increased proliferation of vascular smooth muscle cells, as well as the increase in collagen synthesis and its deposition in the vascular wall [29]. The vascular toxicity of homocysteine has been linked to the impaired endothelial production of nitric oxide, and the overproduction of oxidative radicals that induce intimal injury [30]. In addition, homocysteine occurring as a thiolactone, a highly reactive product of homocysteine oxidation, modulates gene expression in pathways that are important for vascular homeostasis and pathways that are linked to the hyperhomocysteinemia-induced endothelial dysfunction and vascular disease [31,32]. Homocysteine auto-oxidation has been shown to generate the superoxide anion radicals that support the oxidation of low-density lipoproteins [33]. In view of these pro-atherogenic effects, hyperhomocysteinemia is considered a very important modifiable risk factor for cardiovascular diseases [34]. Previously, an in vivo study demonstrated the correlation between homocysteine blood concentration and the concentration and expression of cytokines and proinflammatory and proatherogenic factors, such as Il-6 [35], Il-1β, Il-8, Il-12 [10], Il-10 [11] and RANTES [12,13]. However, the evidence of a direct link between hyperhomocysteinemia and the induction of an immune response leading to the exacerbation or suppression of cytokine secretion by PBMNCs remains unclear. Our in vitro study demonstrated that homocysteine had the ability to stimulate PBMNCs to cytokine secretion. Homocysteine stimulates the secretion of interleukin 1β, interleukin 6 and RANTES chemokines. These studies provide direct evidence for the close relationship between high homocysteine levels in the blood and the intensity of inflammation. A kind of a logical continuity can be the result of the homocysteine-induced IL-10 secretion. The expression of Il-10 was clearly suppressed in the presence of this aminoacid. Considering the antiatherogenic profile of IL-10, the presence of this cytokine in low-concentrations is considered to be a favourable condition for the progression of atherosclerosis and can be even referred to as a proatherogenic factor. Holven et al., [9] demonstrated a hyperhomocysteinemia-dependent decrease in the IL-10 secretion. The mechanism of this is not fully understood. It was documented that homocysteine, especially at concentrations equal to or higher than 100 μmol/L, induced the mRNA and protein expressions of inflammatory cytokines [10] and that homocysteine-relevant transcriptional signaling was dependent on class 1 transcriptional factors, such as HSF, MEF2, NF-AT, NF-κB, KLF4. The hyperhomocysteinemia not only induced the secretion of Il-1β, Il-6, RANTES by monocytes, but also promoted inflammation via the TNFα induction, and subsequent MEF2 and NF-κB transactivation [36]. In addition, hyperhomocysteinemia reduced the expression of anti-inflammatory genes, i.e., PPARγ and PPARα in human monocytes [37]. Il-10 secretion by monocytes was closely related to the activation of PPARγ [38]. The last observation seems to be confirmed by our experiment result, which showed that the Il-10 concentration in homocysteine-containing medium was significantly lower than in the control supernatant.

The results of our own investigation into the impact of high homocysteine concentrations on the immune response by the secretion of interleukin 12 by PBMNCs have proved surprising, as no significant homocysteine action was observed in this regard, although the production of this cytokine was regulated by the NF-κB activity. However, the NF-κBp65/p50 linked to Il-12p40 production was the most shared NF-κB in most of PBMNCs (T-cells, monocytes, NKT-cells) [39], while NF-κBp50/c-Rel linked to Il-12p35 production was present mainly in B-cells [40] which made up only a few percent of all PBMCs. This suggests that homocysteine was mainly responsible for Il-12p40 production by PBMCs, much less than for Il-12p35 and this disproportionately affected any significant increase in Il-12p70. Moreover, the Flex Set test used for the study was focused on the Il-12p70 evaluation, but not on p40 subunit.

## 4. Materials and Methods

### 4.1. Examined Subjects

The peripheral blood samples from ten healthy volunteers (6 males, 4 females, 42–50 years of age, BMI—21.5–26.6, blood pressure below 140/90 mmHg, non-smokers, alcohol consumption < 70 g per week) were collected by the puncture of the peripheral vein into a tube containing EDTA (Sarsted, Nümbrecht, Germany). The volunteers were identified as the outpatients of general practitioners in Poznan (Poland). Before they were eligible for the study, a complete physical examination and laboratory evaluation were performed. Subjects with a history of infections, chronic or acute inflammatory diseases, organ dysfunction (kidney, liver, heart renal failure, thyroid gland disease or anaemia), as well as persons treated with folic acid, vitamin B6 or vitamin B12, were excluded from this study.

### 4.2. The Isolation of Lymphocytes and Peripheral Blood Monocytes

PBMNCs were isolated from the peripheral blood and taken into a tube containing EDTA. The blood was diluted with buffered saline solution of PBS (Sigma-Aldrich, St. Louis, MO, USA) at the ratio of 1:1, laminated in a centrifuge tube on the Lympho-Paque reagent (Genaxxon bioscience, Ulm, Germany) and centrifuged for 20 min, 400× *g* at room temperature. The cells were washed with a PBS solution and suspended in an RPMI-1640 (Merck KGaA, Darmstad, Germany) culture medium for further analysis. The final concentration of the cell suspension in the medium was 2 × 10^6^ mL. The density of the cell suspension was assessed in the Thoma hematocytic chamber. Cell viability was checked with the trypan blue exclusion test (Trypan Blue solution, Sigma-Aldrich, St. Louis, MO, USA).

### 4.3. Cell Cultures

72 h cultures were carried out on sterile 96-well cell culture plates under 5% CO_2_ in air at 37 °C. 200 μL of the cell suspension was placed in the wells, and, subsequently, a mitogenic factor consisting of phytohemagglutinin (PHA, Sigma-Aldrich, St. Louis, MO, USA) at the final concentration of a 2.5 μg/mL and homocysteine solutions (Sigma-Aldrich) were added. A freshly prepared solution of homocysteine at the final concentration of 185 μmol/L was used in the study. Our decision to use homocysteine at the concentration of 185 umol/L was based on the results of the pilot study, in addition to the results published by Su et al. [10]. The controls were wells containing either the culture medium or the cell suspension alone, plus wells containing the cell suspension with the addition of homocysteine-free PHA of equivalent volume.

The final composition of the contents of the wells prepared for individual subjects was as follows:

200 μL of the cell suspension, 10μL of PHA, 40 μL of a 185μmol/L homocysteine solution, 240 μL of the cell suspension, 10 μL of PHA, 240 μL of the cell suspension, 250 μL of the RPMI medium.

Immediately after the completion of the culture period, the samples were centrifuged and the supernatant was stored at −80 °C for further assays.

All procedures performed in studies involving human participants were in accordance with the ethical standards of the institutional and/or national research committee (the Ethics Committee of the Poznan University of Medical Sciences nr 1286/06 (7 December 2006) and 1069/07 (6 December 2007) and all participants provided written informed consent.

The methodology determined the concentration of Il-1β, Il-6, Il-10, Il-12 and RANTES in cell supernatants

The quantitative analysis of Il-1β, Il-6, Il-10, Il-12 and RANTES was based on the flow cytofluorimetry method using Human Il-1β, Il-6, Il-10, Il-12, RANTES Flex Set tests (Becton Dickinson Bioscience, Franklin Lakes, NJ, USA). The analysis included adding beads covered with antibodies (capture beads) directed against the cytokines to the cell culture supernatant. Each population of the capture beads was characterized by a well-defined position in the matrix diagram of FL3 and FL4 fluorescence channels, which made it possible to distinguish them. Subsequently, a solution of antibodies conjugated to the fluorescence-based indicator PE, i.e., phycoerythrin, was added. Both types of antibodies were specific for other binding sites of the test molecules, and the fluorescence of the phycoerythrin-labelled antibodies was directly proportional to the concentration of the cytokines tested. The last stage was measuring the fluorescence of the indicator conjugated to the antibody. At the same time, serial dilution of the standard solutions were made and subjected to the same procedure as the analysed samples. The analysis of the serial dilution of standard solutions produced a standard curve, which was used to calculate the concentration of test molecules in the sample. After the flow cytofluorimetry was performed, graphs showing the number of the cytokines examined were produced. The data were then analysed using FCAP ArrayTM Software (Becton Dickinson) to obtain cytokine concentration values.

The intra- and inter-assay CV was:

Il-1β CV-10%, 8%, 7% and CV-3%, 2%, 4%, Il-6 CV-6%, 6%, 8% and CV-3%, 3%, 2%, Il-10 CV-6%, 10%, 11% and CV-6%, 2%, 3%, Il-12 CV-4%, 2%, 3% and CV-2%, 5%, 3%, RANTES CV-10%, 7%, 3% and CV-4%, 5%, 3%.

### 4.4. Statistical Analysis

Results are expressed as mean ± SD. A comparison of variables between the examined groups was performed using ANOVA (parametric distribution) with Sheffe’s post hoc test. All statistical analyses were performed using Statistica 6.0 (StatSoft, Inc., Dell, Round Rock, TX, USA). A *p*-value ≤ 0.05 was considered statistically significant.

## 5. Conclusions

To summarise the results of our research, homocysteine should be considered a proatherogenic factor, which plays a significant role in the progression of atherosclerosis by directly triggering the inflammatory response by increasing the secretion of Il-1β, Il-6, RANTES and attenuating IL-10 secretion by PBMNCs.

## Figures and Tables

**Figure 1 molecules-26-06671-f001:**
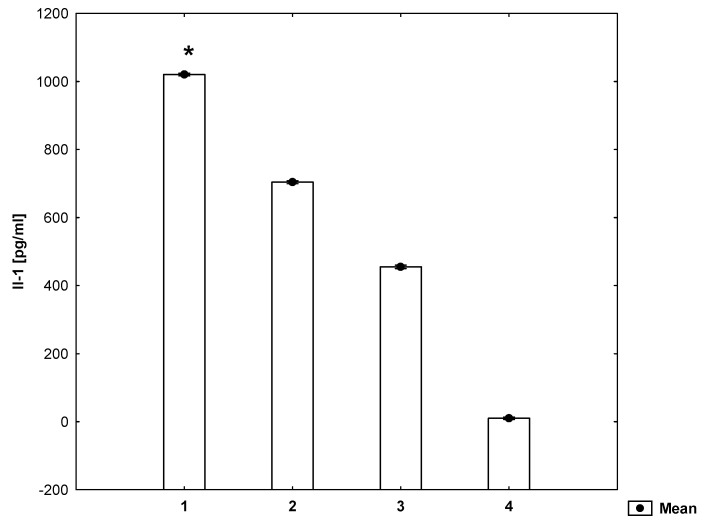
Concentrations of Il-1β in the cell culture supernatant (ANOVA). 1—cell suspension + mitogen + homocysteine, 2—cell suspension + mitogen, 3—cell suspension, 4—medium. * statistically significant difference compared to every other group for *p* ≤ 0.05.

**Figure 2 molecules-26-06671-f002:**
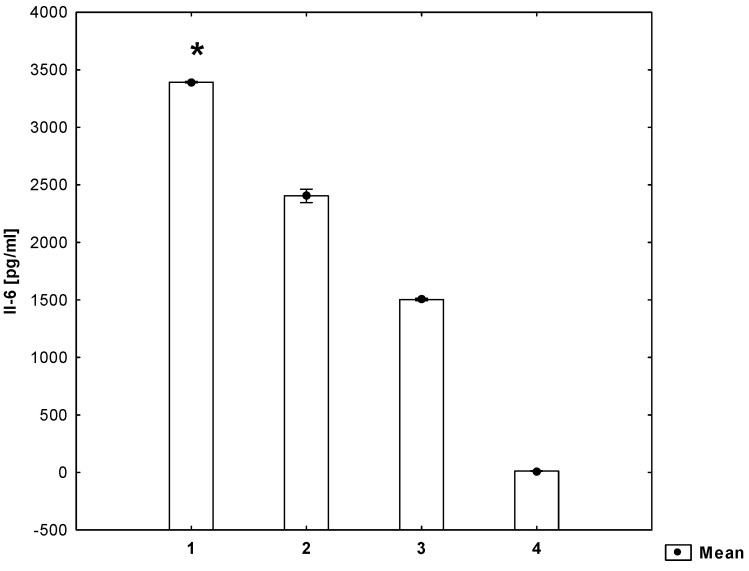
Concentrations of Il-6 in the cell culture supernatant (ANOVA). 1—cell suspension + mitogen + homocysteine, 2—cell suspension + mitogen, 3—cell suspension, 4—medium. * statistically significant difference compared to every other group for *p* ≤ 0.05.

**Figure 3 molecules-26-06671-f003:**
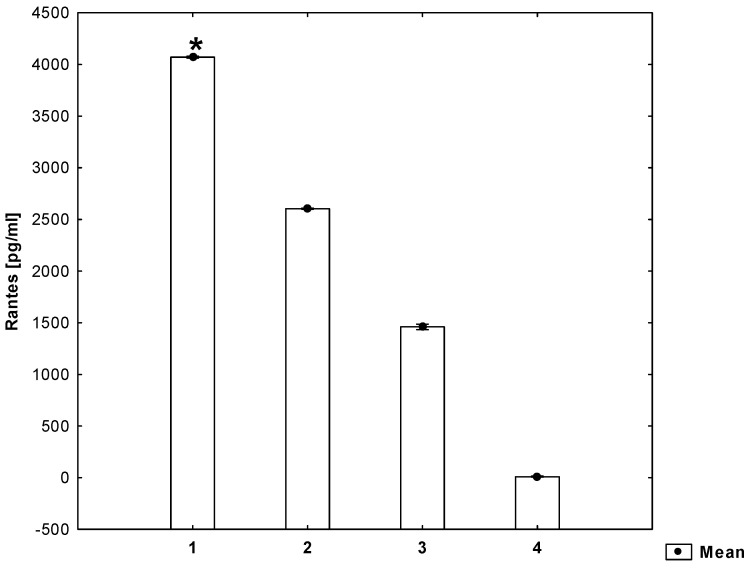
Concentrations of RANTES in the cell culture supernatant (ANOVA). 1—cell suspension + mitogen + homocysteine, 2—cell suspension + mitogen, 3—cell suspension, 4—medium. * statistically significant difference compared to every other group for *p* ≤ 0.05.

**Figure 4 molecules-26-06671-f004:**
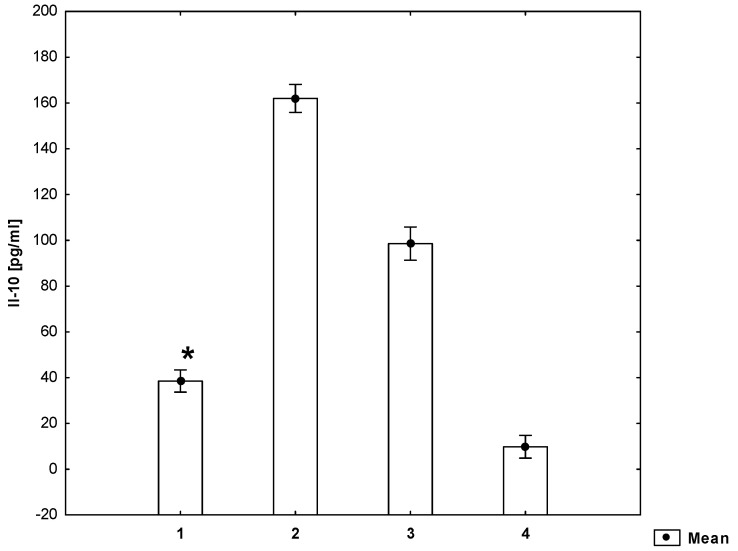
Concentrations of Il-10 in the cell culture supernatant (ANOVA). 1—cell suspension + mitogen + homocysteine, 2—cell suspension + mitogen, 3—cell suspension, 4—medium. * statistically significant difference compared to every other group for *p* ≤ 0.05.

**Figure 5 molecules-26-06671-f005:**
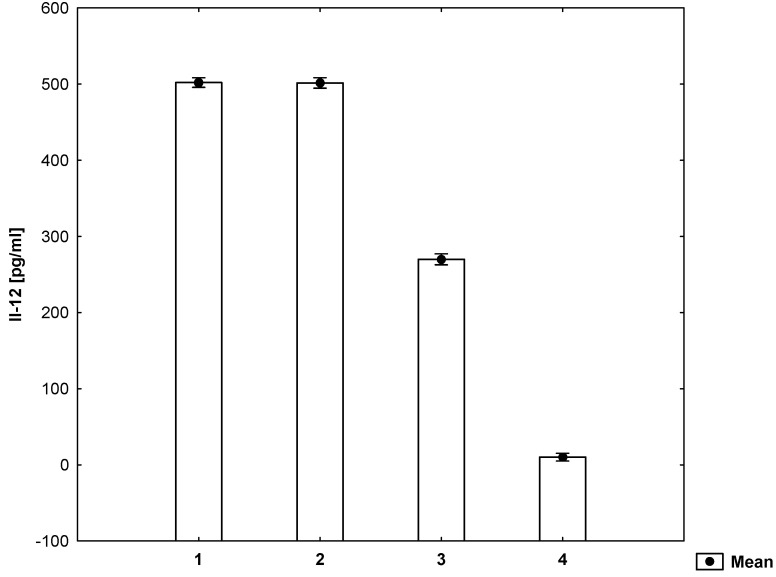
Concentrations of Il-12 in the cell culture supernatant (ANOVA). 1—cell suspension + mitogen + homocysteine, 2—cell suspension + mitogen, 3—cell suspension, 4—medium.

**Figure 6 molecules-26-06671-f006:**
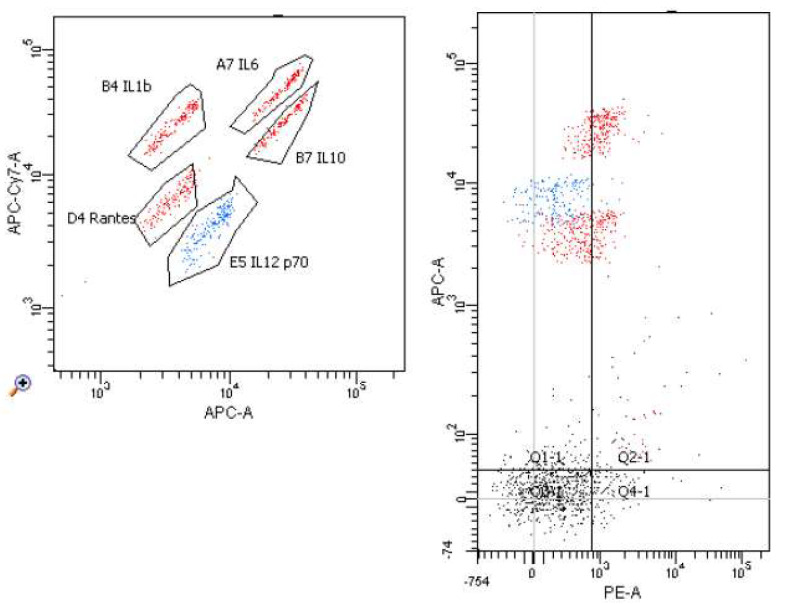
Il-1β, Il-6, Il-12, Il-10 and RANTES expression in cell suspension with mitogen and homocysteine (Flow Cytometry raw data for a single patient).

**Figure 7 molecules-26-06671-f007:**
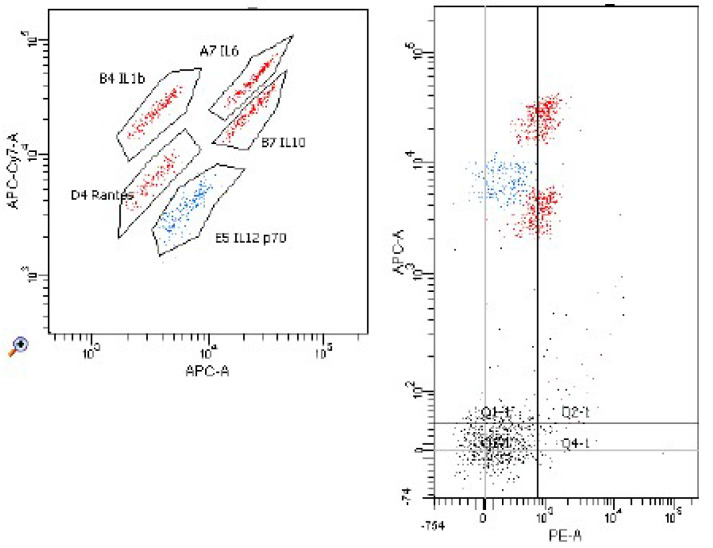
Il-1β, Il-6, Il-12, Il-10 and RANTES expression in cell suspension with mitogen (Flow Cytometry raw data for a single patient).

**Figure 8 molecules-26-06671-f008:**
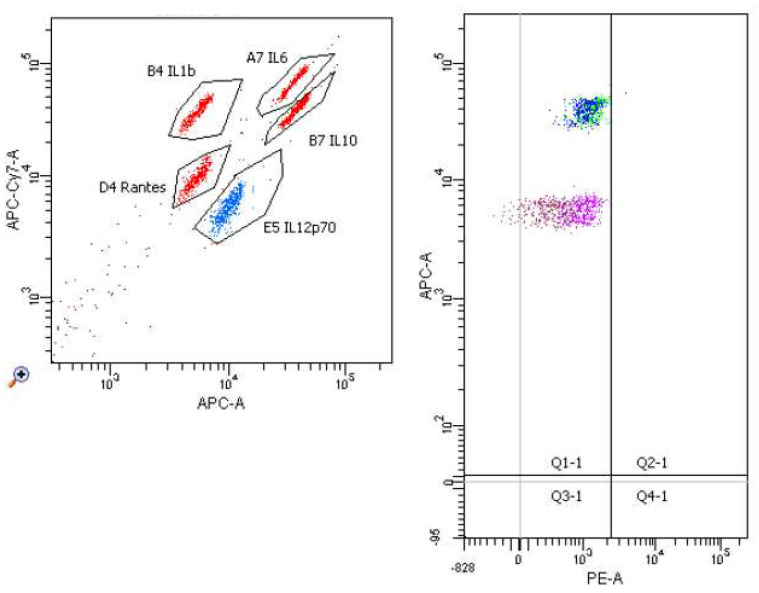
Il-1β, Il-6, Il-12, Il-10 and RANTES expression in cell suspension (Flow Cytometry raw data for a single patient).

**Figure 9 molecules-26-06671-f009:**
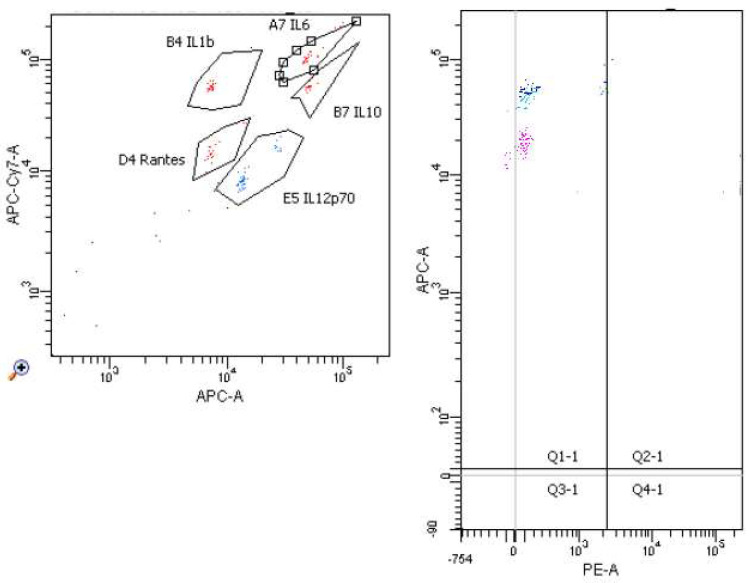
Il-1β, Il-6, Il-12, Il-10 and RANTES expressions in medium (Flow Cytometry raw data for a single patient).

## Data Availability

The data presented in this study are available on request from the corresponding authors.

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
