# Peer review of "The Effect of Homocysteine on the Secretion of Il-1β, Il-6, Il-10, Il-12 and RANTES by Peripheral Blood Mononuclear Cells—An In Vitro Study"

_molecules, 2021, doi:10.3390/molecules26216671_

Round 1
Reviewer 1 Report
- For statistics method, please obtain the result of normality test. My experience about wet lab data usually reject normality, that means, we should not use parametric statistic method. However, the authors use parametric statistic, and I thick this data is necessary. If the statistics method is not correct, the p value is not reliable.
- Please obtain obtain the significance about the flowcytometry result, including the author's interpretation.
Author Response
- For statistics method, please obtain the result of normality test. My experience about wet lab data usually reject normality, that means, we should not use parametric statistic method. However, the authors use parametric statistic, and I thick this data is necessary. If the statistics method is not correct, the p value is not reliable.
Author’s answer:
All results of normality test are attached.
2. Please obtain obtain the significance about the flowcytometry result, including the author's interpretation.
Author’s answer:
All data presented changes in cytokines concentrations were obtained using flow cytometry method, exactly using Human Il-1β, Il-6, Il-10, Il-12, RANTES Flex Set tests (Becton Dickinson Bioscience, New Jersey, USA). These results are presented on Figures 1-5 with statistical significance marked and described in Results chapter.
Figures 6-9 present examples of raw flow cytometry data for each condition used in the experiment (i.g. for the cell suspension with mitogen or for cell suspension with mitogen and homocysteine) received for a single patient. These figures should confirm that each interleukin measured by flow cytometry was well gated and identified, and this way-well measured .

Reviewer 2 Report
In the manuscript entitled “ The effect of homocysteine on the secretion of Il-1β, Il-6, Il-10, Il-12 and RANTES by peripheral blood mononuclear cells - an in vitro study” Authors analyzed association between homocysteine levels and secretion of proinflammatory cytokines. Evidence showed significant positive correlations of homocysteine concentration levels with Il-1β, Il-6 , RANTES and negative with IL-10 whereas surprisingly Authors do not reveled any significant association with IL-12.
The manuscript is well structured and provides a thorough analysis of correlations between homocysteine levels and selected cytokines. It considered suitable for publication in Molecules, I have only some minor concerns
- Abstract (lines 20-21) “..by increasing the secretion of proinflammatory cytokines secreted by PBMNCs, such as Il-1β, Il-6, Il-10 and RANTES. “ and Conclusion (lines 247-248) “triggering of the inflammatory response by increasing the secretion of Il-1β, Il-6, Il-10 and RANTES by PBMNCs” according to data obtained by Authors IL-10 levels decreased, it’s probably a typing error, please correct.
- Homocysteine should stimulate formation of both NF-κBp65/p50 (linked to IL-12p40 production) and NF-κBp50/c-Rel (linked to IL-12p35 production) so question is why there is no biologically active p70? Please elaborate.
- Lines 166-168 : “It may be explained by the properties the Flex Set test focused on the Il-12p70 evaluation, but not on p40 subunit. The last may have resulted in a low specificity of the test Flex Set used.” If test is focused on p70 but not to p40 subunit but specificity of test is really low, still each subunit would be recognized and the result would be false positive. It t is inconsistent with the obtained by Authors data, please correct it.
Author Response
- Abstract (lines 20-21) “..by increasing the secretion of proinflammatory cytokines secreted by PBMNCs, such as Il-1β, Il-6, Il-10 and RANTES. “ and Conclusion (lines 247-248) “triggering of the inflammatory response by increasing the secretion of Il-1β, Il-6, Il-10 and RANTES by PBMNCs” according to data obtained by Authors IL-10 levels decreased, it’s probably a typing error, please correct.
Author’s answer:
Yes, it is corrected in the last version of the manuscript.
2. Homocysteine should stimulate formation of both NF-κBp65/p50 (linked to IL-12p40 production) and NF-κBp50/c-Rel (linked to IL-12p35 production) so question is why there is no biologically active p70? Please elaborate.
Author’s answer:
NF-κBp65/p50 linked to Il-12p40 production is the most shared NF-κB in most of PBMNCs (T-cells, monocytes, NKT-cells) [39], while NF-κBp50/c-Rel linked to Il-12p35 production is present mainly in B-cells [40] which make up only a few percent of all PBMCs. It suggests that homocysteine is mainly responsible for Il-12p40 production by PBMCs, much less for Il-12p35 and this disproportion affected any significant increase in Il-12p70.
3. Lines 166-168 : “It may be explained by the properties the Flex Set test focused on the Il-12p70 evaluation, but not on p40 subunit. The last may have resulted in a low specificity of the test Flex Set used.” If test is focused on p70 but not to p40 subunit but specificity of test is really low, still each subunit would be recognized and the result would be false positive. It is inconsistent with the obtained by Authors data, please correct it.
Author’s answer:
Yes, it was corrected in the last version of the manuscript.
- Hayden, M.S.; Ghosh, S. Signaling to NF-kappaB. Genes Dev. 2004, 18, 2195-224. doi: 10.1101/gad.1228704.
- Miyamoto, S.; Maki, M.; Schmitt, M.J.; Hatanaka, M.; Verma, I.M. Tumor necrosis factor alpha-induced phosphorylation of I kappa B alpha is a signal for its degradation but not dissociation from NF-kappa B. Proc. Natl. Acad. Sci. 1994, 91, 12740-12744.